# Comprehensive Review on the Clinical Relevance of Long Non-Coding RNAs in Cutaneous Melanoma

**DOI:** 10.3390/ijms22031166

**Published:** 2021-01-25

**Authors:** Vincenzo De Falco, Stefania Napolitano, Daniela Esposito, Luigi Pio Guerrera, Davide Ciardiello, Luigi Formisano, Teresa Troiani

**Affiliations:** 1Department of Precision Medicine, Università della Campania “Luigi Vanvitelli”, 80131 Napoli, Italy; vincenzo.defalco2@unicampania.it (V.D.F.); stefania.napolitano@unicampania.it (S.N.); luigipio.guerrera@unicampania.it (L.P.G.); davide.ciardiello@unicampania.it (D.C.); 2Department of Clinical Medicine and Surgery, University of Naples “Federico II”, 80131 Napoli, Italy; daniela.esposito1989@gmail.com (D.E.); luigi.formisano1@unina.it (L.F.)

**Keywords:** lncRNA, melanoma, skin cancer, biomarkers, cancer therapy

## Abstract

Cutaneous melanoma is considered a rare tumor, although it is one of the most common cancers in young adults and its incidence has risen in the last decades. Targeted therapy, with BRAF and MEK inhibitors, and immunotherapy revolutionized the treatment of metastatic melanoma but there is still a considerable percentage of patients with primary or acquired resistance to these therapies. Recently, oncology researchers directed their attention at the role of long non-coding RNAs (lncRNAs) in different types of cancers, including melanoma. lncRNAs are RNA transcripts, initially considered “junk sequences”, that have been proven to have a crucial role in the fine regulation of physiological and pathological processes of different tissues. Furthermore, they are more expressed in tumors than protein-coding genes, constituting perfect candidates either as biomarkers (diagnostic, prognostic, predictive) or as therapeutic targets. In this work, we reviewed all the literature available for lncRNA in melanoma, elucidating all the potential roles in this tumor.

## 1. Introduction

Cutaneous melanoma is the deadliest among the skin cancers and its incidence has increased over the last 30 years [1]. It is completely resectable in most cases but thicker melanomas, especially those with lymph-node involvement, have high risk of relapse. Approximately 5–10% occur in a familial context, associated with mutation in cyclin-dependent kinase inhibitor 2A (*CDKN2A*) gene or, less frequently, in alteration of other genes like breast cancer 1-associated protein (*BAP1*), telomerase reverse transcriptase (*TERT*) or protection of telomeres 1 (*POT1*) [2]. However, most of them are sporadic with the highest prevalence of somatic mutations among cancers [3] and the major risk factor is exposure to the ultraviolet radiation (UVR) of the sun, which causes formation of pyrimidine dimers, photoproducts, oxidative stress and inflammation [2]. Almost 70% of melanomas show an alteration in the mitogen-activated protein kinase (MAPK) signaling pathway, which leads to increased proliferation, invasion and migration. In particular, *NRAS* mutations represent 10–25% of cases, and mutation in the exon 15, codon 600, of the proto-oncogene *BRAF* (*BRAF V600*) accounts for 50% of the alterations, constituting a negative prognostic factor [4]. Recent clinical trials have demonstrated a statistically significant increase of both progression-free survival (PFS) and overall survival (OS) in *BRAF V600* melanoma patients treated with BRAF and MEK inhibitors [5]. Furthermore, cutaneous melanoma interestingly shows a considerable response to immune checkpoint inhibitors (ICIs). Indeed, both antibodies against cytotoxic T-lymphocyte antigen-4 (CTLA-4) and programmed death-1 (PD-1) exhibited high efficacy in terms of PFS and OS both in metastatic and adjuvant settings [6]. Immunotherapy and targeted therapy have dramatically changed the management of resectable and unresectable melanoma in the last years, increasing five-year survival in metastatic patients from 27% up to 52% [6]. Nevertheless, primary or acquired resistance to therapies still remain the main issue for melanoma patients. Moreover, the lack of predictive biomarkers is an unmet need in this disease.

Long non-coding RNAs (lncRNAs)—transcripts considered only “transcriptional junk” until a few years ago—are transcripts longer than 200 nucleotides and constitute a class of regulatory RNAs involved in gene expression regulation. While it has been assessed that lncRNAs lack protein-coding potential, some of them include short open reading frames (sORFs), resulting in the translation of stable and functional peptides [7]. lncRNAs are usually transcribed by RNA polymerase II and then they are processed with 5′ capping, polyadenylation, alternative splicing or RNA editing. According to their genomic position related to neighbor protein-coding genes, lncRNAs can be classified in (1) “intergenic” if they do not intersect any protein-coding genes, (2) “intronic” if they arise within introns of protein coding genes, (3) “sense” if they are transcribed from the same strand of protein-coding genes and overlap their exons and/or introns, (4) “antisense” if they are transcribed from the antisense strand of protein coding genes and (5) “bidirectional” if they are transcribed near (<1000 base pairs) the promoter region of a protein-coding gene but in the opposite direction [8]. lncRNAs can interact with DNA elements, with RNAs and proteins in multiple configurations, mainly exerting their functions (1) as “scaffold”, binding together proteins to constitute a complex; (2) as “decoy”, moving away DNA-binding proteins from target DNA; (3) as “guide”, recruiting chromatin modifiers to DNA to either activate or repress it; (4) as “enhancer”, exerting an enhancer-like function through chromosome looping that spreads lncRNAs effects. [8]. Furthermore, unlike messenger RNAs (mRNAs), lncRNAs display poor sequence conservation among different species, although lncRNAs’ promoters display higher or similar conservation to promoters of protein-coding genes or preserve functional conserved domains through secondary structures [9,10,11]. Transcriptome analysis and in vitro/in vivo assays have progressively highlighted that lncRNAs play a crucial role both in physiological and pathological cellular processes, including cancer cell transformation [9]. Moreover, although lncRNA cell abundance is generally lower than for mRNAs, they exhibit a higher cell/tissue specificity, suggesting they are both new potential biomarkers and therapeutic targets [9].

In this review, we will discuss all the relevant findings about lncRNAs herein described in melanoma prognosis, diagnosis and therapies. Moreover, we will review the mechanisms of action of the most important lncRNAs involved in the tumorigenesis and in the metastatic process of melanoma and the important progresses in the detection of lncRNAs in body fluids as novel clinical biomarkers.

## 2. BRAF Mutation Drives Specific lncRNA Expression 

In recent years, the high throughput RNA sequencing (RNA-Seq) approach has provided a revolutionary method for systematic discovery of transcription units associated with cancer development, including melanoma. Several studies, gaining advantages from large-scale cancer genomics projects, such as The Cancer Genome Atlas (TCGA) and the International Cancer Genome Consortium (ICGC), have revealed a differential and peculiar transcriptional signature between melanoma and normal tissues [12]. Iyer et al., using a compendium of 7256 RNA-Seq libraries from 27 tissues and cancer types, identified almost 60,000 lncRNAs, 339 of which were associated with melanoma [13]. Khaitan et al., using a RNA microarray, found 77 lncRNAs differentially expressed in the BRAF-mutated melanoma cell line (WM1552C) compared to melanocytes, and identified Sprouty RTK Signaling Antagonist 4-Intronic Transcript 1 (*SPRY4-IT1*), involved in melanoma pathogenesis (see Section 5) [14]. Considering the huge percentage of melanomas associated to BRAFV600E mutation, several studies focused their attention on the correlated lncRNAs. 

Flockhart and colleagues verified the expression profiles of BRAF-driven cancer tissue vs. non-BRAF-mutant melanocytes. Interestingly, the analysis revealed that BRAFV600E mutation drives not only the expression of protein-coding genes (1027 specific protein-coding transcripts), but also a peculiar non-coding RNA signature. Thirty-nine annotated lncRNAs and 70 de novo assembled intergenic lncRNAs were identified as differentially expressed between the two groups, including one lncRNA that was named BRAF-regulated lncRNA 1 (*BANCR*) [15]. Several studies have demonstrated its aberrant expression in melanoma patients and melanoma cell lines and have verified its role as a negative prognostic factor. *BANCR* silencing impairs MAPK signaling pathway, inhibiting tumor growth and migration. *BANCR* knockdown, indeed, is able to inhibit melanoma cell migration by upregulating the chemokine CXCL11, and to impair melanoma cell proliferation by modulating ERK1/2 and JNK (MAPK pathway). Interestingly, MEK1/2 or JNK pharmacological inhibition combined with BANCR silencing synergistically affects the proliferative and migration capability of melanoma cells [16]. Mechanistically, *BANCR* can act as competitive endogenous RNA (ceRNA) for miR-204, inducing in turn the activation of the Notch2 pathway [17].

Similarly, a close correlation between BRAFV600E-mutated samples and *RMEL3* lncRNA was described by Goedert and colleagues. *RMEL3* knockdown in BRAF-mutated cell lines results in a significant impairment of cell growth and survival. Moreover, *RMEL3* silencing leads to deregulation of proteins involved in MAPK and PI3K pathways (decreased b-Raf and p-AKT levels and increased tumor suppressor PTEN levels), as well as of cell cycle and apoptosis regulators (decreased p-RB and cyclin-B1 levels and increased p-21 and p-27 levels) [18]. Finally, BRAFV600E melanoma cells treated with BRAF or MEK inhibitors significantly decreased *RMEL3* expression levels, suggesting *RMEL3* as a downstream effector of ERK signaling and as a promising candidate for pharmacological target [19].

Nevertheless, little is known about melanoma lncRNAs induced by driver mutations different from BRAFV600E. ZEB1 antisense RNA 1 (*ZEB1-AS*) is a recent lncRNA whose expression levels have been associate not only to BRAF mutation, but also to RAS mutations in metastatic melanoma samples according to TCGA data [20]. However, RNA-Seq analysis in melanoma samples stratified for a mutational landscape rather than the tumor stage could significantly improve the knowledge about the lncRNA signature in melanoma, which could be definitely expand to all the tumor types harboring the same mutational state. 

## 3. Prognostic Value of lncRNAs in Cutaneous Melanoma and Their Role in Therapy Resistance

The identification of a risk-classifying lncRNA signature for melanoma patients represents a useful prognostic biomarker to improve clinical outcomes, to design employed therapies and to boost the overall patients’ survival [21,22]. Using gene expression profiles in melanoma patients from the TCGA data, Yang et al. have identified a signature of six lncRNAs to stratify patients, comparing their expression in stage I-II vs. stage III-IV melanoma samples. Furthermore, the analysis of target genes of the identified lncRNAs revealed their regulatory role in MAPK signaling, immune and inflammation-related pathways, the neurotrophin and focal adhesion pathways—closely associated with cancer progression—suggesting the six-lncRNA signature as a potential biomarker to improve patient survival [23]. Focusing on lncRNAs that can act as ceRNAs to regulate the expression of mRNAs through miRNA regulation, a 7-lncRNA model to predict the overall survival of melanoma patients was identified, although the biological significance of these lncRNAs is still unclear [24]. Similarly, trying to reconstruct an lncRNA-miRNA-mRNA network based on a different expression between primary melanoma and benign nevi tissue samples, Zhu et al. identified three lncRNAs closely related to tumorigenesis in melanoma. In particular, *LINC00943*, *LINC00261* and *MALAT1*—related to various malignant tumors [25,26,27]—were identified and validated in an independent cohort of patients as predictive molecules for melanoma treatment and as potential therapeutic targets [28]. Metastasis-associated lung adenocarcinoma transcript 1 (*MALAT1*) and also urothelial carcinoma-associated 1 (*UCA1*) have been further investigated in a cohort of independent patients, confirming the over-expression of both lncRNA in later stage of metastatic melanomas compared to primary tumors, suggesting a putative role of these lncRNAs in promoting metastasis [29]. Finally, a combination of 3 lncRNA-based risk score and five clinicopathologic factors were used to build a nomogram to predict three-, five-, and 10-year overall survival (OS) in patients with cutaneous melanoma. This recently developed method could help to define an individualized program of treatment for melanoma patients [30].

The identification of a specific lncRNA signature could provide an easy-to-apply method for acquiring risk information in order to stratify melanoma patients and to suggest the best therapeutic option. Since lncRNA signatures may be affected by driver gene mutations (as suggested in Section 2), in silico devices may be implemented considering the mutational state of patients to describe lncRNA prediction signatures suitable to specific subsets of patients.

Considering the pivotal role of lncRNAs in regulating crucial genes involved in tumor progression, lncRNA involvement in resistance to target therapies has been evaluated as well. In BRAF V600E melanoma cell lines (A375), a wide genome-scale activation screening using the Clustered Regularly Interspaced Short Palindromic Repeats Cas9 (CRISPR-Cas9) technique has identified an lncRNA—named EQTN MOB3B IFNK C9orf72 enhancer RNA I (*EMICERI*)—as a prediction factor of resistance to targeted therapy. In particular, the transcriptional activation of *EMICERI* regulates nearby genes and confers vemurafenib resistance, via upregulation of MOB3B and subsequent stimulation of the Hippo signaling pathway [31]. The well-studied long noncoding RNA X-inactive specific transcript (XIST), instead, was described as upregulated in melanoma tissues and cell lines compared to normal tissues and, interestingly, its down-modulation was associated with restoring sensitivity to oxaliplatin in oxaliplatin-resistant cells [32]. Moreover, the over-expression of the lncRNA Taurine-Upregulated Gene 1 (*TUG1*) in melanoma tissues and cell lines correlates with poor prognosis. It acts as an oncogene sponging miR-129-5p and inducing cell growth and invasion. miR-129-5p sequestration by TUG1 leads to the over-expression of astrocyte-elevated gene-1 (AEG-1), a protein involved in the PI3K/AKT pathway, the WNT signaling pathway and chemo-resistance to cisplatin and 5-fluorouracil [33].

The cytoplasmic intergenic lncRNA *MIRAT* (MAPK inhibitor resistance-associated transcript), instead, has been described as significantly over-expressed in melanoma cells, carrying NRAS or BRAF mutations, resistant to small molecule inhibitors of the MAPK cascade. On the other hand, the specific silencing of the *MIRAT* lncRNA in such resistant cells restores sensitivity to MEK inhibitors. Authors suggested that *MIRAT* lncRNA binds to IQGAP1—boosting MEK to ERK signaling and hyper-activation in different cancer types—and stabilizes it, thus pushing MAPK signaling [34].

In conclusion, lncRNA aberrant expression on the one hand could guide the acquisition of resistant phenotypes, but on the other hand could also result from such resistance. Hence, the molecular and mechanistical knowledge of the mRNA/lncRNA network could prevent the acquisition of resistance and improve the specific targeted therapies.

## 4. lncRNA Mechanisms of Action Described in Cutaneous Melanoma

lncRNAs modulate the expression of protein coding genes by acting at various levels (e.g., chromatin remodelling, transcription, RNA processing and stability, protein translation). Several lncRNAs act at their site of transcription and impact genes localized in the surroundings or on the same chromosome (cis-acting lncRNAs) and others affect distant genes on the same chromosome or on other chromosomes (trans-acting lncRNAs). Their roles are strongly related to subcellular localization, acting at transcriptional and post-transcriptional levels in the nucleus and at a post-transcriptional level in the cytosol. Functional studies have progressively revealed that lncRNAs are crucial players in several hallmarks of cancer (e.g., proliferation, evasion of cell death and metastasis), enabling an expansion of the definition of oncogenes and tumor suppressors to lncRNAs.

### 4.1. Tumor-Suppressor lncRNAs

Cancer susceptibility candidate 2 *(CASC2)*, *LINC00961*, *LINC00459* and maternally expressed gene 3 (*MEG3)* have been described as tumor-suppressor lncRNAs in melanoma. Their expression is lower in melanoma cell lines and tissues compared to controls and explicates their function by acting as ceRNA for microRNAs, thus by sequestering miRNAs from their targets and modulating miRNA-mediated post-transcriptional silencing. In particular, *CASC2* exerts its tumor-suppressor function sponging miR-18a-5p and consequently promoting RUNX1, a tumor growth inhibitor in different cancers [35]. Moreover, CASC2 facilitates the expression of PLXNC1 by binding to miR-181a, thus resulting in anti-proliferative effects [36]. *LINC00961* inhibits cell proliferation and promotes apoptosis sponging miR-367, in turn regulating the expression of phosphate and tension homolog (PTEN) and its downstream pathway [37]. Long intergenic non-protein-coding RNA 459 (*LINC00459)* was recently discovered with a microarray assay on a cohort of melanoma samples and pigmented nevus samples. Its expression levels are lower in tumor tissues compared to the pigmented nevus tissues and, surprisingly, the median overall survival in the *LINC00459* low-expression group is significantly lower than in the high-expression group. Moreover, in vitro and in vivo assays defined its involvement in modulating cell viability, cell cycle, apoptosis and migration/invasion acting as a ceRNA and regulating the miR-218/DKK3 pathway (known to be involved in cancer hallmarks [38]. Finally, *MEG3* suppresses proliferation and shows the pro-apoptotic function in melanoma as well. While several studies have revealed that the *MEG3* function is mediated, at least in part, by the activation of the p53/MDM2 axis [39], its function in melanoma as ceRNA for miR-499-5p and miR-21 has been recently demonstrated by regulating CYLD and E-cadherin expressions, respectively [40,41].

Similarly, Nuclear Factor-Kappa B Interacting lncRNA (*NKILA*) and growth-arrest specific 5 (*GAS5)* exert pro-apoptotic functions in melanoma by different mechanisms. *NKILA* acts at a post-transcriptional level by interacting and interfering with IκB phosphorylation, which leads to NF-κB activation (promoting the anti-apoptotic pathway) [42]. Several mechanisms of action, instead, have been proposed for *GAS5*, including decoy or miRNA sponge activity. As a decoy, *GAS5* interacts with the DNA-responsive elements of the glucocorticoid receptor, preventing its binding to the DNA, thereby blocking the transcription of target genes involved in anti-apoptotic processes [43]. However, over-expression of *GAS5* in melanoma cell lines induces a decreased expression of matrix metalloproteinases (MMPs), specifically involved in extracellular matrix (ECM) degradation. This event results in an impaired capability of cells to migrate and invade, ascribing to GAS5 enhancement a putative therapeutic value [44].

Disrupted in Renal Carcinoma 3 (*DIRC3*) lncRNA is, instead, a MITF-SOX10-regulated nuclear lncRNA, exerting its tumor-suppressor function by cis-acting and, in particular, by chromatin remodeling. It activates expression of its neighboring Insulin Like Growth Factor Binding Protein 5 (*IGFBP5*) tumor suppressor gene, impacting the expression of its target genes [45].

### 4.2. Oncogene lncRNAs

*SAMMSON* (survival-associated mitochondrial melanoma-specific oncogenic non-coding RNA)—located downstream of the specific oncogene of melanoma microphthalmia-associated transcription factor (MITF)—is an oncogenic lncRNA expressed in more than 90% of human melanomas. In contrast, it is only poorly detectable in normal human melanocytes and in melanoma lesions in radial growth phase [46]. *SAMMSON* expression is regulated by SOX10, a transcription factor located upstream of the *SAMMSON* transcription start site (TSS), and is involved in melanoma malignancy by enhancing mitochondrial metabolism. It has been described, indeed, that *SAMMSON* interacts with p32—required for 12S ribosomal RNA processing—leading to regulation of the mitochondrial metabolism [46]. Furthermore, *SAMMSON* silencing using locked nucleic acid (LNA)-modified antisense oligonucleotides (GapmeRs) resulted in a marked reduction of cells’ clonogenic ability and in a massive cell death (independently of their NRAS, BRAF or TP53 status), as well as in an enhanced cytotoxic effects of BRAF and MEK inhibitors in melanoma cell lines and patient-derived xenograft (PDX) [46]. Surprisingly, SAMMSON knockdown synergized with BRAF and MEK inhibitors also in cells with acquired resistance to BRAF inhibitors, probably due to the addiction of resistant cells to mitochondria oxidative phosphorylation. These results revealed its oncogenic activity in melanoma and its contribution as a new effective and tissue-specific therapeutic target.

Antisense non-coding RNA in the INK4 locus (*ANRIL*) is a well-studied antisense lncRNA, transcribed from the locus INK4b-ARF-INK4a, encoding for three tumor-suppressor proteins, p15, p14 and p16, respectively. Its oncogenic capacity is exerted by polycomb repressor complex (PRC1 and PRC2) recruitment, which induces gene repression both to the promoters of its neighboring genes (cis activity) and to distant targets through ALU sequence (trans activity) [47]. Its over-expression—documented in different cancer types including melanoma—bypasses growth suppression processes and promotes tumor phenotype. Consistently, *ANRIL* silencing activates the expression of *INK4a* and *INK4b*, thus significantly reducing the tumorigenesis of melanoma [48].

Likewise, steroid receptor RNA activator 1-like non-coding RNA (*SLNCR1*) exerts its oncogenic function acting as a scaffold. *SLNCR1* robust expression has been associated with worse overall melanoma survival; Schmidt et al. demonstrated that *SLNCR1* bind to brain-specific homeobox protein 3a (Brn3a) and the androgen receptor (AR), constituting a complex with high affinity for the proximal *MMP9* promoter. This results in a transcriptional upregulation of *MMP9* that enhances melanoma cell invasion [49].

Acting as a guide, focally amplified lncRNA on chromosome 1 (*FALEC*) recruits EZH2, an RNA binding protein, to p21 (inhibitor of cyclin-dependent kinase) and induces its epigenetic silencing. Accordingly, *FALEC* silencing produces growth inhibition and cell cycle arrest and enhances apoptosis, suggesting its role as oncogenic lncRNA [50]. In addition, lymph node metastasis-associated transcript 1 (*LNMAT1*) acts as a guide for EZH2 to suppress cell adhesion molecule 1 (CADM1) expression, which in turn, as member of the cell adhesion molecule family, acts as a tumor suppressor inhibiting matrix metalloproteinases involved in ECM degradation [51].

Similar to some tumor suppressor lncRNAs, non-coding RNA activated by DNA damage (*NORAD*) explicates it oncogenic activity sponging miR-205, a tumor suppressor in melanoma. MiR-205 has inhibitory effects on migration and invasion of melanoma cell lines by targeting EGLN2 (an invasion-associated gene and thus inducing endoplasmic reticulum (ER) stress. Consistently, stable short hairpin-mediated knockdown of NORAD lncRNA inhibits ER stress and thus interferes with the migration and the invasion of melanoma cell lines [52]. Forkhead box D3-Antisense RNA 1 (*FOXD3-AS1*), instead, seems to promote proliferation, invasion and migration in melanoma cell lines through the miR-325/MAP3K2 axis. MAP3K2 is, indeed, involved in numerous pathways like MAPK signaling, β-catenin pathway and Hedgehog. Hence, interfering with MAP3K2 activation by modulating *FOXD3-AS1* could revert proliferation, invasion and migration of melanoma cells [53].

Finally, the testis-associated highly conserved oncogenic long non-coding RNA(*THOR*) behaves as an oncogene by binding to IGF2BPs, so stabilizing at post transcriptional levels their target mRNAs. In this scenario, *THOR* may act as an oncogene lncRNA in melanoma (where it was found upregulated compared to control samples) promoting the cancer phenotype, as suggested by zebrafish knockout models that defect in melanoma onset [54].

## 5. lncRNA Detection in Body Fluids of Melanoma Patients as Novel Clinical Application

lncRNA levels have been described as de-regulated in cancer tissues compared to nonmalignant cells. The latest evidence shows that lncRNAs—as well as microRNAs—can be secreted into the extracellular space in macrovesicles or exosomes, complexed with proteins or high-density lipoproteins, making them stable and preventing their degradation by endonucleases. Consequently, circulating lncRNAs can be detected in body fluids, including serum, plasma, urine and saliva, ascribing a new crucial role in prognosis and diagnosis for lncRNAs [55]. Hence, evaluation of lncRNA expression levels in body fluids of cancer patients—instead of classical tumor biopsies of tumor tissues—represents a non-invasive and safe method helpful for clinical applications.

The homeobox transcript antisense intergenic RNA (*HOTAIR*) is one of the 231 ncRNAs associated with human HOX loci and was one of the first discovered lncRNAs that regulate gene expression. It is upregulated in melanoma cells and tissues, with progressively higher expression from benign nevi to primary tumors and to metastatic lesions. It has been also identified in some intratumoral lymphocytes and in the serum of metastatic melanoma patients [56]. In melanoma cell lines, it promotes proliferation, invasion and migration and the Epithelial-to-Mesenchimal Transition (EMT). *HOTAIR* mechanisms of action are still debated and several hypotheses have been supported. Mainly, it has been shown to act as a key regulator of chromatin states by binding to the specific chromatin modification complex polycomb repressive complex 2 (PRC2), thereby recruiting and affecting PRC2 occupancy on target genes [57]. Furthermore, *HOTAIR* interacts with lysine-specific histone demethylase 1A (LSD1), which exerts its function in epigenetic regulation by modulating the methylation of lysine 4 of histone H3 (H3K4) and in the silencing of target genes by costituting a multiprotein complex via activation of the RE1-silencing transcription factor (REST) and CoREST [58]. Finally, *HOTAIR* can act also as ceRNA for miR-152-3p, which in turn regulates the *MET* mRNA, inducing the activation of the downstream PI3K/AKT/mTOR-signaling pathway [59]. Cantile and colleagues verified the expression of the lncRNA *HOTAIR* in melanoma lesions compared to healthy tissue as well as its circulating levels in the blood of patients. Interestingly, they found *HOTAIR* over-expression in tissues in correlation with the advancement of disease stage and, interestingly, they also detected increased *HOTAIR* levels in the serum of metastatic patients, suggesting its potential role in melanoma metastatic progression and as a monitor for therapeutic response [56]. Similarly, the long intergenic non-protein coding RNA 1638 (*LINC01638*)—whose over-expression enhances the proliferation of melanoma cells—is over-expressed in tumor biopsies and plasma samples of melanoma patients compared to patients with benign skin lesions and healthy controls. Interestingly, serial liquid biopsies from stage I-IIIA of melanoma patients revealed significantly higher levels of *LINC01638* in those who exhibited local recurrence than in patients without recurrence, suggesting its involvement in the recurrence process, although the mechanisms by which it occurs still need to be elucidated [60].

Sprouty RTK signaling intagonist 4-intronic transcript 1 (*SPRY4-IT1*, also known as *SPRIGHTLY*) is a cytoplasmic lncRNA transcribed from an intron of the sprouty 4 (*SPRY4*) gene and is highly expressed in melanoma cells [14]. Down-modulation of SPRY4-IT1 by siRNA-mediated knockdown inhibited invasion and proliferation and induced apoptosis of melanoma cells, suggesting an important role for this lncRNA in melanoma onset and progression. In particular, it seems to be involved in lipid metabolism modulating cellular concentrations of lipin 2 substrates, including phosphatidate [61]. Furthermore, ectopic over-expression of SPRY4-IT1 in vitro was associated with downregulation of tumor suppressor gene DPPIV/CD26 and consequent upregulation of genes involved in cell proliferation, including MAPK-extracellular signal-regulated kinase (ERK) 1/2 [62]. Notably, high levels of *SPRY4-IT1* have been detected in plasma samples of melanoma patients and are closely associated with tumor sites and tumor stages [63]. Patients with high *SPRY4-IT1* expression, indeed, showed shorter survival than those with low *SPRY4-IT1* expression, regardless of patients’ sex, age and histologic type.

Finally, serum levels of Plasmacytoma variant translocation 1 (*PVT1*) were measured in melanoma patients in the study of Chen and colleagues [64]. Several studies found PVT1 over-expression in melanoma cell lines and in melanoma tissues compared to melanocytes, identifying its role as a poor prognostic factor. In vitro studies revealed that PVT1 knockdown inhibits proliferation, induces cell cycle arrest at the G0/G1 phase and enhances the apoptotic events in melanoma cell lines [65]. Moreover, PVT1, by binding to EZH2, enhances the activity of miR-200c, involved in cancer progression and in regulation of EMT [66]. PVT1 serum expression levels—defined by quantitative reverse transcription polymerase chain reaction (qRT-PCR)—revealed a significant increase in patients with melanoma (higher expression in later stages compared to early ones) compared with control group. This study suggested PVT1 serum detection as a novel biomarker for melanoma early diagnosis and could have clinical relevance in melanoma as either a diagnostic serum biomarker in early stages or as a monitor of disease in advanced disease [64].

Instead focusing on BRAF-mutant metastatic melanoma patients Kolenda and colleagues evaluated the associations between the expression levels of lncRNAs and patients’ responses to vemurafenib treatment. lncRNA plasma expression in melanoma patients treated with vemurafenib vs. healthy donors was quantified by qRT-PCR and, interestingly, Zeb2NAT, Zfas1, 7SL and AIR were identified as significantly associated with progression [67].

In order to establish lncRNA detection in body fluids as clinical routine analysis, the promising results obtained herein need to be implemented. Trials with established clinical end points of disease progression and/or survival, which define specific treatments based on lncRNA results collected up to now, will definitely describe the advantages and reliability of monitoring lncRNAs in body fluids to improve tumor progression follow up and response to therapy.

## 6. Conclusions

lncRNAs represent finer and more specific regulators of cellular processes than protein-coding genes in many cancers. In cutaneous melanoma, they could be used as diagnostic tools, as prognostic and prediction biomarkers and as pharmacological targets. Indeed, as shown in this review, different studies demonstrated the role of lncRNAs into tumorigenesis and progression of cutaneous melanoma. However, the majority of these studies must be confirmed in in vivo studies and in a broader cohort of patients. Moreover, for the large number and variety of lncRNAs involved in melanoma progression (Table 1), we think that targeting only one of them is not effective enough and does not work on all patients.

**Table 1 ijms-22-01166-t001:** Main lncRNAs with a putative function in melanoma.

lncRNA	Role in Melanoma	Functional Mechanism	Expression in Melanoma	Direct Target	Pathway Regulated	Ref.
**ANRIL**	Oncogene	Scaffold	Upregulated	PRC1, PRC2	INK4B-ARF-INK4A	[47]
**BANCR**	Oncogene	Decoy	Upregulated	miR-204	EKR1/2, JNK; NOTCH2	[15,16,17]
**CAR10**	Oncogene	Decoy	Upregulated	miR-125b-5p	RAB3D	[68]
**CASC2**	Tumor suppressor	Decoy	Downregulated	miR-18a-5p miR-181a	RUNX1; PLXNC1	[35,36]
**CASC15**	Oncogene	Guide	Upregulated	EZH2	PDCD4	[69,70]
**CCAT1**	Oncogene	Decoy	Upregulated	miR-33a	HIF-1α	[71]
**CRNDE**	Oncogene	Decoy	Upregulated	miR-205	CCL18	[72]
**DIRC3**	Dual role	Guide	/	/	IGFBP5	[45]
**FALEC**	Oncogene	Guide	Upregulated	EZH2	p21	[50]
**FOXD3-AS1**	Oncogene	Decoy	Upregulated	miR-325	MAP3K2	[53]
**GAS5**	Tumor suppressor	/	Downregulated	/	MMP2, MMP9	[44]
**H19**	Oncogene	/ Decoy	Upregulated	/ miR-106a-5p	MMP2, MMP3, VIM, CDH2, MST1R, CDH1; E2F3	[73,74]
**HEIH**	Oncogene	Guide	Upregulated	EZH2	miR-200b/a/429	[75]
**HOTAIR**	Oncogene	Scaffold, Decoy	Upregulated	PRC2, LSD1; miR-152-3p	MET, PI3K/AKT/mTOR	[57,59]
**ILF3-AS1**	Oncogene	Guide	Upregulated	EZH2	ILF3, miR-200b/a/429	[76]
**LHFPL3-AS1**	Oncogene	Decoy	Upregulated	miR-181	Bcl-2	[77]
**LINC00173**	Oncogene	Decoy	Upregulated	miR-493	IRS4	[78]
**LINC00459**	Tumor suppressor	Decoy	Downregulated	miR-218	DKK3	[38]
**LINC00518**	Oncogene	Decoy	Upregulated	miR-204-5p	AP1S2	[79]
**LINC00520**	Oncogene	Decoy	Upregulated	miR-125b-5p	EIF5A2	[80]
**LINC00961**	Tumor suppressor	Decoy	Downregulated	miR-367	PTEN	[37]
**LINC01638**	Oncogene	/	Upregulated	/	/	[60]
**LLME23**	Oncogene	Guide	Upregulated	PSF	RAB23	[81]
**lncRNA-ATB**	Oncogene	Decoy	Upregulated	miR-590-5p	YAP	[82]
**LNMAT1**	Oncogene	Guide	Upregulated	EZH2	CADM1	[51]
**MALAT1**	Oncogene	Decoy	Upregulated	miR-22; miR-34a; miR-140; miR-608	MMP14, SNAIL; c-Myc, MET; Slug, ADAM10; HOXC4	[83,84,85,86]
**MEG3**	Tumor suppressor	Decoy	Downregulated	miR-499-5p; miR-21	CYLD, E-cadherin, N-cadherin, CyclinD1	[40,41]
**MHENCR**	Oncogene	Decoy	Upregulated	miR-425; miR-489	IGF1; SPIN1, PI3K/AKT	[87]
**MIR31HG**	Oncogene	/	Upregulated	/	p16INK4A	[88]
**MIRAT**	Oncogene	/	Upregulated	IQGAP1	MAPK pathway	[34]
**NEAT1**	Oncogene	Decoy	Upregulated	miR-495-3p; miR-23a-3p	E2F3; KLF3	[89,90]
**NKILA**	Tumor suppressor	/	Downregulated	/	NF-kβ	[42]
**NORAD**	Oncogene	Decoy	Upregulated	miR-205	EGLN2	[52]
**OIP5-AS1**	Oncogene	Decoy	Upregulated	miR-217	GLS	[91]
**PANDAR**	Oncogene	/	Upregulated	/	NF-YA	[92]
**PVT1**	Oncogene	Guide	Upregulated	EZH2	miR-200c	[64,65,66]
**RMEL3**	Oncogene	/	Upregulated	/	MAPK and PI3K pathways	[18,19]
**SAMMSON**	Oncogene	/	Upregulated	p32	Mitochondria metabolism	[46]
**SLNCR1**	Oncogene	Scaffold	Upregulated	Brn3a, AR	MMP9	[49]
**SNHG5**	Oncogene	Decoy	Upregulated	miR-155; miR-26a-5p	TRPC3	[93,94,95]
**SPRY4-IT1**	Oncogene	/	Upregulated	/	Lipid metabolism; DPPIV/CD26, MAPK pathway	[14,61,62]
**THOR**	Oncogene	/	Upregulated	/	IGF2BP pathway	[54]
**TSLNC8**				PP1α	MAPK pathway; response to BRAF inhibitor	[96]
**TTN-AS1**	Oncogene	Decoy	Upregulated	TTN	/	[97]
**TUG1**	Oncogene	Decoy	Upregulated	miR-129-5p	AEG-1, PI3K/AKT, WNT	[33]
**UCA1**	Oncogene	Decoy	Upregulated	miR-507; miR-28-5p	FOXM1; HOXB3	[98,99]
**ZEB1-AS1**	Oncogene	/	Upregulated	/	/	[100,101]
**ZFAS1**	Oncogene	Decoy	Upregulated	miR-150-5p	RAB9A	[102]
**ZFPM2-AS1**	Oncogene	Decoy	Upregulated	miR-650	NOTCH1	[103]

## Data Availability

Not applicable.

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
