# Peer review of "Comprehensive Review on the Clinical Relevance of Long Non-Coding RNAs in Cutaneous Melanoma"

_ijms, 2021, doi:10.3390/ijms22031166_

Round 1
Reviewer 1 Report
In this attempted review from Falco et al., the authors tried to compile an overview of long non-coding RNAs that might be important in cutaneous melanoma. I think the review is in bad shape and as it is of now just a list of titles from papers that have lncRNAs associated or have been found in cutaneous melanoma. It has no heart or structure and lacks inspiration that makes it interesting for readers. All sections starting in number 5 are just way too short and superficial leaving the reader with more questions than answers and from the looks just summarize the title of most cited publications.There is no connection or flow in the text.
The conclusion is short, boring and doesn't give any interesting aspect what the author thinks or how a future outlook in this field could look like. The outlook that targeting a single lncRNA is not enough is also very confusing. With all the different mutations that can occur leading to melanoma development, it might be expected that different lncRNAs could play different roles in different mutations. For example, the authors could check whether different mutations bear different lncRNAs or always the same thus connecting different publications/mutations together. The Conclusion also doesn't go into detail how those lncRNAs identified could be used as biomarker or even recapitulate if they are already used. How can lncRNAs be pharmacological targets? Do you mean pathways that are activated by lncRNAs could be targeted making melanoma cells more susceptible to treatment with inhibitors? The authors need to discuss each claim they make in the discussion.
In conclusion I would reject the review in present form.
Minor comments:
Line 51: "...that cannot be translated into proteins..." is not a true statement. Most lncRNAs do not code for proteins (hence their classification as non-coding) but it has been shown that some produce small peptides that are relevant.
Line 54: please rephrase the whole sentence. "LncRNAs belong to a class call non-coding RNAs which include....." Then in the next sentence you can describe the properties of lncRNAs and add that lncRNAs longer than 200nt in lenght to distinguish them from miRNAs in that sentence.
Line 58: First of all they are classified by their genomic location for now, not can be and change into to in the genome
Line 59: No, not composed of intronic sequences. How is an intronic sequence composed. The better way: "'intronic" when they originate from an intron"
Line 60: No, please only list the real locations. There might be some that overlap exons of protein coding genes but in no review so far they have been classified that way
Line 62: starting
Line 58-63: The classification is missing enhancer lncRNAs
Line 71: You have to give the reader explanations to statements that you make, why is there poor sequence conservation, because there is less evolutionary pressure on lncRNAs
Line 72: what are unique temporal and spatial expression patterns, this sentence is a good example of an empty sentence giving no information what so ever. In fact lncRNAs are so interesting because they are more specific to a certain cell type (or cancer for that matter) than mRNAs.
Line 121: What does this sentence mean?
Author Response
Thank you for the precious suggestions. The review was completely revised with the help of two experts in RNA field (added as authors). We are thankful if you will revise again our work in order to publish it.
Reviewer 2 Report
This review is dealing with an important health problem melanoma, which frequency is increasing over years.
LnRNAs have been considered for a long time as junk RNAs dedicated to degradation, while the recent years have revealed their functional importance in the regulation of protein gene expression and cellular processes as well as in the evolution of numerous pathologies.
The authors are cancer specialists and they made a big effort to collect a large amount of information on the link between lncRNAs and melanoma from published data.
Although, the thematic is exploding and that numerous papers are published in the field, there is already about five reviews published this year on lncRNAs and melanoma, which limits somehow the impact of the proposed review.
One major criticism is that the manuscript is lacking a scientific organization. The very long paragraph on putative mechanisms of action of lncRNAs in melanoma consists in an enumeration of a huge number of individual observations described in the different papers published in the field.
The authors should try to define different domains of lncRNAs activity in order to architecture this long paragraph. They should then try to point out the most important general mechanisms by which lncRNAs may influence melanoma development and progression.
In its present organization of this paragraph one can wonder if a large table gathering all the data would not be as informative as the present text.
Another point is that as the authors are specialists of cancer but not of lncRNAs some of the assumptions are sometime too strong or not correct.
As an example, the authors claim that the main function of the ANRIL lncRNA is to negatively regulate transcription in cis of INK4B‐ARF‐INK4A through association with polycomb repressor complexes (PRC1 and PRC2), while it is quite clear now that in addition to this function in cis, ANRIL exerts several other functions in trans.
At several places in the text, the authors conclude that an up regulation of the level of a given lncRNA suggests that it plays a role in the pathology. There may be increases of lncRNA levels without dedicated function, which are just resulting from a change in the global balance of transcription at certain genomic loci. The authors must be more cautious in their conclusions.
The review needs a very strong revision. The advices are :
- to improve the organization of the review, in order that it becomes scientifically more attractive,
- to be quite more cautious in the statements, the best would be to include one lncRNA specialist among the authors, who will contribute to a better understanding and evaluation of the proposed putative roles of lncRNAs and their possible clinical applications,
- in addition, English has to be seriously improved at several places
Author Response

(The authors gave the same response as above.)

Round 2
Reviewer 1 Report
I was going through the MS and there are still a lot of small errors which I highlighted but probably didn’t get them all. So please check again. The MS is now in good shape to be published.
Author Response
Dear reviewer, all the minor revisions suggested by you were accomplished. Thank you for your precious help.
